# Nanorod-like Structure of ZnO Nanoparticles and Zn_8_O_8_ Clusters Using 4-Dimethylamino Benzaldehyde Liquid to Study the Physicochemical and Antimicrobial Properties of Pathogenic Bacteria

**DOI:** 10.3390/nano13010166

**Published:** 2022-12-30

**Authors:** Sivalingam Ramesh, C. Karthikeyan, A. S. Hajahameed, N. Afsar, Arumugam Sivasamy, Young-Jun Lee, Joo-Hyung Kim, Heung Soo Kim

**Affiliations:** 1Department of Mechanical, Robotics and Energy Engineering, Dongguk University—Seoul, Pil-dong, Jung-gu, Seoul 04620, Republic of Korea; 2Departemnt of Chemical and Biochemical Engineering, Dongguk University—Seoul, Pildong-ro 1 gil, Jung-gu, Seoul 04620, Republic of Korea; 3PG and Research Department of Physics, Jamal Mohamed College (Affiliated to Bharathidasan University), Tiruchirappalli 620020, Tamil Nadu, India; 4PG & Research Department of Chemistry, L. N. Government College, Ponneri 601204, Tamil Nadu, India; 5Catalysis Science Laboratory, Chemical Engineering Area, Central Leather Research Institute (CLRI-CSIR), Adyar, Chennai 600020, Tamil Nadu, India; 6Department of Mechanical Engineering, Inha University, Inha-ro 100, Namgu, Incheon 22212, Republic of Korea

**Keywords:** ZnO, Zn_8_O_8_ clusters, organic additives, optical properties, first-order hyperpolarizability, thermal properties, antimicrobial properties

## Abstract

To study their physicochemical and antimicrobial properties, zinc oxide nanoparticles were synthesized using a simple chemical route and 4-dimethylaminobenzaldehyde (4DB) as an organic additive. ZnO nanoparticles were characterized with XRD analysis, which confirmed the presence of a hexagonal wurtzite structure with different crystalline sizes. The SEM morphology of the synthesized nanoparticles confirmed the presence of nanorods in both modifications of ZnO nanoparticles. EDS analysis proved the chemical composition of the synthesized samples via different chemical approaches. In addition, the optical absorption results indicated that the use of 4DB increased the band gap energy of the synthesized nanoparticles. The synthesized Zn_8_O_8_ and Zn_8_O_8_:4DB clusters were subjected to HOMO–LUMO analysis, and their ionization energy (I), electron affinity (A), global hardness (η), chemical potential (σ), global electrophilicity index (ω), dipole moment (μ), polarizability (α_tot_), first-order hyperpolarizability (β_tot_), and other thermodynamic properties were determined. Furthermore, the antimicrobial properties of the ZnO nanoparticles were studied against G+ (*S. aureus* and *B. subtilis*) and G− (*K. pneumoniae* and *E. coli*) bacteria in a nutrient agar according to guidelines of the Clinical and Laboratory Standards Institute (CLSI).

## 1. Introduction

Nanotechnology research is widely used in materials science for various applications, such as innovative fabric compounds, agricultural production, medicinal techniques, and food processing methods. The synthesis and characterization of materials in the nanometer scale region of 1–100 nm have been explored for various chemical modifications [1,2,3]. This new technology focuses on the synthesis of controlled nanomaterials to enhance their physicochemical and biological properties. Biosensors, gas sensors, electrochemical sensors, supercapacitors, nanomedicine, and bionanotechnology have all made extensive use of nanoscale materials [4,5]. With the use of various chemical modifications, these metal oxides have been widely used for their antimicrobial activity against various pathogenic bacteria [6,7]. The improved antimicrobial properties are due to the metal oxides’ important role in oxidation and the inhibition of reactive oxygen species in biological conditions. Zinc oxide (ZnO) is considered a next-generation material for chemical or biosensors because of its piezo activity, transparency, and conductivity [8,9]. ZnO is one of the most promising materials for short-wavelength light-emitting devices and technological applications due to its band gap energy of 3.37 eV [10] and high exciton-binding energy of 60 meV. The large excitonic-binding energy of ZnO nanoparticles (NPs) and the extreme stability of excitons at room temperature (RT) enable devices to function at low threshold voltages. The optical properties of ZnO NPs are more interesting, since the confinement of charge carriers in the restricted volume of the small particles can lead to the enlargement of the band gap [10]. Recently, several new routes, such as organometallic precursors [11], sol–gel synthesis [12], precipitation [13], and solvothermal and hydrothermal methods [14,15], have been developed for the synthesis of ZnO nanostructures. Moreover, ZnO NPs’ morphologies have been controlled by the different reaction conditions and additives used in experiments. Zinc nitrate and hexamethylenetetramine compounds were used to create highly uniform microrods and tubes of ZnO nanoparticles [16,17]. Different morphologies of ZnO nanoparticles have been generated using ethylene diamine for potential applications in sensors and as antimicrobial agents [18,19]. The hexagonal, prismatic, and pancake-like morphologies of ZnO with block copolymers have been used in various applications [17,18,19]. The spiral growth of ZnO plates in aqueous media in the presence of sodium citrate has shown different morphologies for various biological applications [20,21]. Nanosheet-like ZnO morphologies resulted from the use of phosphate ions (PO_4_^3−^) in an aqueous solvent for antimicrobial applications [22,23]. The specific adsorption of functional groups with the structural modification of ZnO NPs has resulted in improved antimicrobial results. Various organic dye modifications of ZnO NPs using eosin Y have been used with an electrochemical process for the purification of different water samples [24,25,26]. Hence, various morphological properties of ZnO NPs on organic molecules have been achieved with different chemical approaches. This may be due to the molecular interactions and the nature of the crystal surface and bonding behavior of nanoparticles in composite systems. In the present study, 4-dimethylaminobenzaldehyde (4DB) was selected as an organic additive in the preparation of ZnO NPs formation of the 4DB metal complexes with various metal ions [27], and sodium acetate, succinic acid, and salicylic acid were used as additives to form different morphologies of ZnO NPs. The organic molecules linked with anionic groups and their morphological properties have various potential applications in the biological field [28]. Additionally, ZnO NPs can be grown at low temperatures by forming soluble zinc amine, which precipitates out as ZnO when the solution temperature is raised [29]. Furthermore, the optical properties of ZnO NPs were found to be improved by optimizing parameters such as pH, precursor concentration, growth time, and temperature. In the present investigation, pure and 4-dimethylaminobenzaldehyde (4DB)-added ZnO NPs were synthesized with a simple chemical route. The ZnO NPs were characterized with X-ray diffraction (XRD), scanning electron microscopy, EDX analysis, UV–Vis NIR, and photoluminescence spectroscopy. These clusters have become an increasingly interesting topic in physics, chemistry, and microscopic materials research for various potential applications. Although numerous theoretical and experimental studies on the structures and properties of bulk ZnO have been performed, only a few investigations have been devoted to (ZnO)_n_ clusters [30,31]. In this work, the HOMO–LUMO energy gap, ionization energy (I), electron affinity (A), global hardness (η), chemical potential (σ), global electrophilicity index (ω), dipole moment (μ), polarizability (α_tot_), first-order hyperpolarizability (β_tot_), and other thermodynamic properties were calculated using the B3LYP/6-311 G (d, p) level of the basic set available in the Gaussian 09W Program. In addition, ZnO NPs can be endowed with unique antimicrobial and antibacterial properties and excellent morphological behaviors via different chemical modifications. The physicochemical properties of ZnO-based metal oxides have been improved in previous studies with environmental influence and different concentrations [32,33,34]. As a result, this study focused on ZnO NPs in 4-dimethylaminobenzaldehyde liquid with varying morphologies and their antimicrobial properties against Gram-positive and Gram-negative bacteria in biological conditions [35].

## 2. Experimental Details

### 2.1. Materials

Analytical-grade zinc nitrate hexahydrate (Zn (NO_3_)_2_·6H_2_O) (98%), sodium hydroxide NaOH (≥98%), and 4-dimethylaminobenzaldehyde (4DB) (99%) were received from Sigma–Aldrich, India.

### 2.2. Synthesis of the ZnO Nanoparticles

To prepare ZnO NPs, a simple chemical one-pot method with 0.1 M of Zn (CH_3_COO)_2_ was used. First, 25 mL of water and a 0.8 M NaOH solution were slowly added to a solution with continuous stirring to obtain a white precipitate. The reaction temperature was maintained at 60 °C while stirring and heating for 4 h. This solution was refluxed at RT for 24 h. A clear solution, which was found to be stable at ambient conditions, was then obtained. After the solution was washed several times with double-distilled water and ethanol, the precipitate was finally dried at 200 °C. Thus, ZnO NPs were obtained for use in various structural, morphological, and antimicrobial studies.

### 2.3. Synthesis of the ZnO:4DB Nanoparticles

4DB-added ZnO NP suspensions were synthesized as described above. A particular amount of 4-dimethylaminobenzaldehyde (4DB) was separately dissolved in a methanol solution. These 4DB solutions were separately added to the respective pure ZnO solutions and stirred for 20 min. The resulting suspension was found to be yellow in color, and after the precipitate was repeatedly rinsed with ethanol and double-distilled water, it was finally dried at 200 °C. Thus, ZnO:4DB nanopowders were acquired for use in physicochemical and antimicrobial studies. Figure 1 shows a schematic of the synthesis of the ZnO:4DB nanoparticles.

### 2.4. Materials Characterization

ZnO NPs were characterized with X-ray diffractometry (X’PERT PRO PANalytical). The diffraction patterns of the ZnO NPs were recorded in the range of 25–80° using a monochromatic light of wavelength 1.54 Å (Cu–Kα). Scanning electron microscopy (SEM) observations were carried out using a JEOL/EO type-JSM-6390, and the elemental compositions were estimated using energy-dispersive X-ray analysis (EDX) (Model; OXFORD). UV–Vis–NIR spectra were recorded in the range of 190–1200 nm using a UV–Vis–NIR spectrometer (Model: Lambda 35). Photoluminescence spectroscopy (PL) measurements were performed with a PerkinElmer fluorescence spectrometer, and the corresponding spectra were recorded in the range of 360–560 nm for the ZnO NP samples.

### 2.5. Computational Analysis

The quantum chemical calculations for the Zn_8_O_8_ and Zn_8_O_8_:4DB clusters were performed using the B3LYP/6-311 G (d, p) level of the Gaussian 09W basic program with the initial geometries of the ground-state structure of the Zn_8_O_8_ and Zn_8_O_8_:4DB clusters without symmetry constraint [36]. 

### 2.6. Antibacterial Assay

Following the guidelines of the Clinical and Laboratory Standards Institute (CLSI), the antibacterial activity of the ZnONPs and ZnO:4DB NPs was investigated using the well diffusion method and tested against G+ (*S. aureus* and *B. subtilis*) and G− (*K. pneumoniae* and *E. coli*) bacteria in a nutrient agar. After inoculation, wells loaded with 1 mg/mL of the test samples were placed on bacteria-seeded well plates using micropipettes. The plates with the bacterial seeds were incubated at 37 °C for 24 h. Then, the inhibition zone was measured. To assess antimicrobial properties, amoxicillin (Hi-Media) was used as a positive control against the G+ and G− bacteria. 

## 3. Results and Discussion

### 3.1. Growth Mechanism

Figure 1 shows a possible mechanism for the formation of the pure and 4DB-added ZnO nanorods. This can be discussed based on both its internal structure and the growth habit of the 4-dimethylaminobenzaldehyde-added ZnO nanorods, which are affected by external factors, such as solution pH, zinc source, the presence of any organic impurities, nucleation condition, and the extent of supersaturation. The structure of ZnO can be described as a number of alternative planes composed of tetrahedrally coordinated O^2−^ and Zn^2+^ ions stacked along the c-axis. ZnO is a polar crystal that has a polar axis, and it possesses positive and negative faces due to the asymmetric distribution of Zn and O atoms along its polar axis. The oppositely charged ions produce positively charged (0001) Zn-polar and negatively charged (000−1) O-polar surfaces, resulting in a normal dipole moment and spontaneous polarization along the c-axis. The ZnO growth along the c-axis occurs along the hexagonal (0001) plane of ZnO nanorods. However, polycrystalline aggregate morphology is present when growing under high driving force conditions [37]. Organic molecules are used to control the growth direction and shape of nanoparticles for different chemical approaches [38]. Furthermore, the aggregation of the nanorods in the 4DB added to ZnO samples in this study was found to be decreased, and the promoted growth along the c-axis resulted in the formation of ZnO nanorods.

### 3.2. Structural Characterization

Figure 2 shows the XRD patterns of the pure and 4DB-added ZnO NPs. The pronounced diffraction peaks corresponding to the (100), (002), (101), (102), (110), (103), (200), (112), (201), (004), and (202) planes clearly showed the crystalline nature of the samples. The standard diffraction peaks showed that the crystal structure of the ZnO NPs was a hexagonal wurtzite structure (space group P63mc, JCPDS data Card No: 36-1451) with preferred orientation along the (101) plane. This was found to be the most stable phase of ZnO. The ‘a’ and ‘c’ lattice constants of the wurtzite structure of ZnO can be calculated using the following relation [39]:(1)$$\frac{1}{d^{2}}=\frac{4}{3}\left(\frac{h^{2}+hk+k^{2}}{a^{2}}\right)+\frac{l^{2}}{c^{2}}$$
with the first-order approximation (*n* = 1) for the (100) plane. The lattice constant ‘*a*’ was calculated with the relation *a* = *λ*/√3sin*θ*, while the lattice constant ‘*c*’ was derived for the (002) plane with the relation *c* = *λ*/sin*θ*. The calculated values of ‘*a*’ and ‘*c*’ were 0.3257 and 0.5217 nm, respectively for the pure ZnO NPs, whereas those values for the 4DB-added ZnO NPs were 0.3255 and 0.5213 nm, respectively. The values showed decreases in the lattice constants due to the effect of the 4DB molecules on ZnO. Table 1 shows the lattice parameter values.
V = 0.866 a^2^ c(2)

The unit cell volumes were calculated using the above relation as 47.9554 and 47.8487 Å for the pure and 4DB-added ZnO NPs, respectively. The unit cell volume also decreased when adding 4DB to the ZnO.

The average crystalline size of the samples was calculated with the Debye–Scherrer relation:(3)$$\mathrm{Average}\;\mathrm{crystal}\;\mathrm{size}\;(D)=\frac{k\lambda }{\beta_{D\;\cos\theta }}$$
where *λ* is the wavelength of the radiation (1.54056 Å for Cu–Kα radiation), *K* is a constant that is equal to 0.94, *β* is the peak width at half-maximum intensity, and *θ* is the peak position. The average crystalline size was reduced from 45 to 43 nm. This clearly demonstrated the presence of nanosized particles in the samples. The reduction in the crystalline size was mainly due to distortion in the host ZnO lattice due to the foreign impurity, i.e., 4DB, which decreased the nucleation and subsequent growth rate of the ZnO NPs.

### 3.3. Morphology and Chemical Composition Analysis

Figure 3a,b shows the surface morphologies of the pure and 4DB-added ZnO NPs, respectively. Many uniform, quality ZnO nanorods with good coverage were formed in the pure ZnO and 4DB-added ZnO NP samples. The average particle sizes of the pure and 4DB-added ZnO NPs were found to be 65 and 71 nm, respectively. An increase in the average particle size of the 4DB-added ZnO was observed. A possible mechanism of the formation of nanorods is discussed in Section 3.1 based on both the internal structure and growth of ZnO NPs. The ZnO and ZnO:4DB NPs were subjected to EDAX elemental analysis, and Figure 3c,d shows their respective EDAX spectral results. Table 2 shows the chemical composition of the synthesized samples annealed at 200 °C. The EDAX analysis revealed that the required phase was present in the samples. The results indicated the formation of high-purity ZnO NPs. For the pure ZnO, the chemical compositional atomic percentage was 61.07 and 38.93% for Zn and O, respectively, whereas for the 4DB-added ZnO, the zinc percentage decreased while the oxygen percentage increased (Table 2).

### 3.4. UV–Vis NIR Spectroscopic Analysis

Figure 4a shows the UV–Vis–NIR optical absorption spectra of the pure and 4DB-added ZnO NPs that were recorded in the range of 190–1100 nm. Absorbance is expected to depend on several factors, such as the optical band gap, oxygen deficiency, surface roughness, and impurity center [40,41]. According to the recorded absorption spectra, absorption peaks were found at 341 nm for the pure ZnO NPs and 338 nm for the 4DB-added ZnO NPs, which can be attributed to the photoexcitation of electron from the valence band to the conduction band. The position of the absorption spectra was observed to shift toward the lower wavelength side for the 4DB-added ZnO NPs. This indicated that the band gap of ZnO NPs increased with the addition of 4DB to the ZnO NPs.

The relation between the absorption coefficient α and the incident photon energy hυ can be written as follows [42]: the band gap increased from 3.26 to 3.285 eV with the addition of 4DB to ZnO. The addition of 4DB to the ZnO NPs induced an increasing band gap compared with the ZnO nanomaterials. Usually, the band gap was increased when the absorption edge shifted toward the lower wavelength side because of the substitution of 4DB into the ZnO surface matrix.

### 3.5. Photoluminescence Spectroscopic Analysis

Figure 5a,b shows the photoluminescence spectra of the as-synthesized ZnO NPs and 4DB:ZnO NP samples recorded with the excitation wavelength of 341 nm. In the photoluminescence (PL) spectra of the ZnO NPs, there are emission bands in the UV and visible regions [40,41,42,43]. In the current study, UV emissions are observed at 394 and 374 nm for the formation of ZnO NPs and 4DB: ZnO NPs, respectively, which corresponded to near band edge (NBE) emission. The pure ZnO NPs and 4DB:ZnO NPs had different emissions that were observed at 414, 458, 478, 493, 520, and 551 nm and at 408, 440, 475, 493, 519, 530, and 560 nm, respectively. The violet emission peak observed at 414 and 408 nm was due to an electron transition from a shallow donor level of natural zinc interstitials to the top level of the valence band [44]. The blue band emissions that were observed at 458, 478, and 493 nm and at 440, 475, and 493 nm may have been due to surface defects in the ZnO NPs corresponding to oxygen vacancies and oxygen interstitial defects, respectively. The green emission peaks observed at 520 and 551 nm and at 519, 530, and 560 nm were assigned to a deep level emission that is usually caused by the presence of an ionized charged state of the defects in zinc oxide. Interestingly, the 4DB-added ZnO NPs only exhibited green emissions, the amount of which was increased compared with ZnO due to the effect of 4-dimethylaminobenzaldehyde.

### 3.6. Computational Studies on the Zn_8_O_8_ and Zn_8_O_8_:DB Clusters

#### 3.6.1. The Structures of the Zn_8_O_8_ and Zn_8_O_8_:4DB Clusters

Figure 6a,b shows the calculated ground configuration of the Zn_8_O_8_ bell-like structure and Zn_8_O_8_:4DB clusters, respectively. The Zn-O bond was found to be primarily ionic, transferring charge from Zn to O atoms. The structural properties of the Zn_8_O_8_ were in good agreement with those reported in the literature [45]. According to the present calculations, the bond lengths of Zn–O and Zn–Zn were found to be 1.40078 and 2.48506 Å, respectively. The bond angle of Zn–O–Zn was calculated as 122.00566 degrees. These values were consistent with previously reported values [46].

#### 3.6.2. HOMO–LUMO Analysis

To evaluate the electronic properties of the Zn_8_O_8_ clusters, we calculated the energy gaps between their HOMO and LUMO states. Electronic absorption corresponds to the transition from the ground state to the first excited state, and it is mainly described by one electron’s excitation from the highest occupied molecular orbital (HOMO) to the lowest unoccupied molecular orbital (LUMO). HOMO represents the ability to donate an electron, and LUMO represents the ability to obtain an electron. Figure 6 shows the calculated energy gaps of the frontier molecular orbitals of the Zn_8_O_8_ and Zn_8_O_8_:4DB clusters. In general, clusters with larger HOMO–LUMO gaps are more chemically inert [47,48,49,50,51,52,53,54] because improving electrons to a high-lying LUMO or extracting electrons from a low-lying HOMO is energetically unfavorable. For the Zn_8_O_8_ cluster, the HOMO energy value was found to be −0.11650 a.u., and the LUMO energy value was found to be −0.08407 a.u. The HOMO–LUMO energy gap value was estimated as −0.03234 a.u. The HOMO was delocalized over all Zn_8_O_8_ clusters in the 4-dimethylaminobenzaldehyde-added Zn_8_O_8_ clusters, while the LUMO was located over the entire benzaldehyde group. Consequently, the HOMO–LUMO transition implied an electron density transfer from the benzaldehyde group. The HOMO energy value was found to be −0.11078 a.u. for the 4-dimethylaminobenzaldehyde-added Zn_8_O_8_ clusters. Their LUMO energy and HOMO–LUMO energy gap were calculated as −0.09186 and −0.01892 a.u., respectively. Compared with Zn_8_O_8_:4DB with Zn_8_O_8_ clusters, the energy values of Zn_8_O_8_ were decreased due to the effect of charge density, which was unequally shared by the cluster in the organic molecules. The calculated self-consistent field (SCF) energy values for the Zn_8_O_8_ and Zn_8_O_8_:4DB clusters were calculated as −14,833.2275 and −15,312.7238 a.u., respectively. In the case of the Zn_8_O_8_-added 4DB clusters, the SCF energy increased due to the effect of 4-dimethylamino benzaldehyde, which increased the surface energy of the Zn_8_O_8_ clusters.

#### 3.6.3. Calculation of Ionization Energy (I), Electron Affinity (A), Global Hardness (η), Chemical Potential (σ), Global Electrophilicity Index (ω) and Dipole Moment (μ)

The ionization energy and electron affinity of clusters are also sensitive quantities that provide fundamental insight into electronic structures. The HOMO–LUMO energy gap reflects the chemical activity of a molecule. Within the framework of SCF MO theory, ionization energy and electron affinity can be expressed through HOMO and LUMO orbital energies as I = −E_HOMO_ and A = −E_LUMO_, respectively. Ionization energy is defined as the energy difference between cationic and the neutral structures. In this study, the cationic ZnO structures were frozen at the optimized neutral clusters because the ‘‘hole’’ left by removing an electron almost instantaneously recombined with an incoming electron from the whole system. Figure 7a shows that the ionization energy decreased in the 4DB-added Zn_8_O_8_ clusters compared with the pure Zn_8_O_8_ due to the effect of the energy gap. The electron affinity was calculated based on the optimized geometry of the neutral cluster. The energy difference between the optimized neutral system and the energy of the negative system at the same geometry was interpreted as the electron affinity of the cluster. The calculated results of the Zn_8_O_8_ clusters are shown in Figure 7a. The electron affinity increased in the Zn_8_O_8_:4DB clusters compared with pure Zn_8_O_8_. The hardness corresponded to the gap between the HOMO and LUMO orbital energies. The larger the HOMO–LUMO energy gap, the harder the molecules. Global hardness is given by η = 1/2(E_LUMO_ − E_HOMO_). The hardness of a chemical system is associated with its stability. The estimation of chemical hardness confirmed the high chemical inertness of the Zn_8_O_8_ clusters (0.0162 a.u.) and the low chemical reactivity of the Zn_8_O_8_:4DB cluster (0.0094 a.u.).

When electron affinity is combined with ionization energy, the electronic chemical potential is given by = 1/2 (E_HOMO_ + E_LUMO_). The electronic chemical potential was mostly found to be the same for the Zn_8_O_8_ and Zn_8_O_8_:4DB clusters. The global electrophilicity index is a measure of energy lowering due to the maximal electron flow between a donor and acceptor. The global electrophilicity index is calculated with the relation ω = μ2/2η. When two molecules react, the one with the higher (lower) electrophilicity index acts as an electrophile (nucleophile). This new reactivity index measures the stabilization in energy when a system acquires an additional electronic charge ∆N from the environment. Electrophilicity is a reactivity descriptor that allows for the quantitative classification of a molecule’s global electrophilic nature on a relative scale. Here, the Zn_8_O_8_:4DB clusters were found to have a more electrophilic nature than the pure Zn_8_O_8_ clusters due to the total energy of the system.

The amount of a molecule’s overall charge determines its polarity. Polarity is therefore a measure of the dipole moment. A chemical bond results from the accumulation of charge density in the binding region to an extent that can sufficiently balance the forces of repulsion. Covalent and ionic charge distributions exhibit radically different chemical and physical properties. Here, the dipole moment of the Zn_8_O_8_ clusters was found to be 7.4989 Debye, whereas that of the Zn_8_O_8_:4DB clusters was found to be 5.3239 Debye. The difference in the dipole moment was due to the effect of charge density, which was unequally shared in the Zn_8_O_8_:4DB clusters. This asymmetry of the charge distribution effect decreased the dipole moment for the Zn_8_O_8_:4DB clusters, as shown in Figure 7b. The calculated values are given in Table 3.

#### 3.6.4. Non-Linear Optical Properties

The mean polarizability (α_tot_), anisotropy of polarizability (∆α), and average value of the first-order hyperpolarizability (β_tot_) of the Zn_8_O_8_ and Zn_8_O_8_:4DB clusters were calculated using the B3LYP/6-311 G(d, p) basis set based on the finite-field approach. In the presence of an applied electric field, the energy of a system is a function of the electric field. First-order hyperpolarizability is a third-rank tensor that can be described by 3 × 3 × 3 matrices. The 27 components of a 3D matrix can be reduced to 10 components due to Kleinman symmetry, and this matrix can be given in the lower tetrahedral format. It is obvious that the lower part of 3 × 3 × 3 matrices is tetrahedral. The components of β are defined as the coefficients in the Taylor series expansion of the energy in the external electric field. When the external electric field is weak and homogeneous, this expansion becomes:E = E^0^ − μ_α_ F_α_ − 1/2α_αβ_ F_α_ F_β_ − 1/6 β_αβγ_ F_α_ F_β_ F_γ_ +……(4)
where E^0^ is the energy of the unperturbed molecules; F_α_ is the field at the region; and μ_α_, α_αβ_ and β_αβγ_ are the components of dipole moments, polarizability, and the first-order hyperpolarizability, respectively. Polarizability (μ_xx,_ μ_xy,_ μ_yy,_ μ_xz,_ μ_yz,_ and μ_zz)_ and first-order hyperpolarizability (β_xxx,_ β_xxy,_ β_xyy,_ β_yyy,_ β_xxz,_ β_xyz,_ β_yyz,_ β_xzz,_ β_yzz,_ and β_zzz_) tensors can be obtained from the output file of Gaussian 09W. However, the α and β values of Gaussian output are in atomic units (a.u.). Therefore, they were converted into electronic units (esu) (for α_tot_, 1 a.u. = 0.1482 × 10^−24^ esu; for β_tot_, 1 a.u. = 8.6393 × 10^−33^ esu) in this study. The mean polarizability (α_tot_), anisotropy of polarizability (∆α), and average value of the first-order hyperpolarizability (β_tot_) can be calculated using Equations (5)–(7), respectively.
(5)$$\alpha_{\mathrm{tot}}=\frac{1}{3}\;(\alpha_{\mathrm{xx}}+\alpha_{\mathrm{yy}}+\alpha_{\mathrm{zz}})$$
(6)$$\Delta \alpha =\frac{1}{\sqrt{2}}{\left[(\alpha_{\mathrm{xx}}-\alpha_{\mathrm{yy}}{)}^{2}+(\alpha_{\mathrm{yy}}-\alpha_{\mathrm{zz}}{)}^{2}+(\alpha_{\mathrm{zz}}-\alpha_{\mathrm{xx}}{)}^{2}+6{\alpha^{2}}_{\mathrm{xz}}+6{\alpha^{2}}_{\mathrm{xy}}+6{\alpha^{2}}_{\mathrm{yz}}\right]}^{1/2}$$
β_tot_ = [(β_xxx_ + β_xyy_ + β_xzz_)^2^ + (β_yyy_ + β_yzz_ + β_yxx_)^2^ + (β_zzz_ + β_zxx_ + β_zyy_)^2^]^1/2^(7)

The calculated parameters as described above are shown in Table 4 and Figure 8 for the Zn_8_O_8_ and Zn_8_O_8_:4DB clusters. Polarizability was used to measure the response of electrons against an external electric field. The weak binding of nuclei to electrons and large delocalization volumes tend to promote polarizability magnitude. Cluster stability can be used to estimate the capacity of the electrons flowing within a cluster, and the delocalization volume is also an important factor that affects polarizability. The calculated polarizability α_ij_ had non-zero values and was dominated by the diagonal components. The total polarizability (α_tot_) was calculated as 40.1698 × 10^−24^ esu and 80.0811 × 10^−24^ esu for the Zn_8_O_8_ and Zn_8_O_8_:4DB clusters, respectively, using B3LYP/6-311 G(d, p) basis set values. The first-order hyperpolarizability was derived from the electron excitations that involve both the ground and excited states. Besides transition energy and the transition dipole moment, the dipole moment difference between the ground and excited states is also important in the calculation of β_tot_. The first-order hyperpolarizability value of the pure Zn_8_O_8_ clusters was decreased compared with that of the Zn_8_O_8_:4DB clusters due to the effect of the total energy. The first-order hyperpolarizability values (β_tot_) of the Zn_8_O_8_ and Zn_8_O_8_:4DB clusters were found to be 157,935.5078 × 10^−33^ esu and 1,199,485.513 × 10^−33^ esu, respectively. The first-order hyperpolarizability β_tot_ was dominated by the longitudinal components β_tot_. The dominance of one component indicated a significant delocalization of charges in its direction.

#### 3.6.5. Thermodynamic Properties

The zero-point vibrational energies (ZPVE), rotational constant, rotation temperature, thermal energy, molar capacity at constant volume, and entropy were calculated with the B3LYP/6-311 G (d, p) basis set. The thermodynamic parameters are given in Table 5. The thermodynamic zero-point vibrational energies, thermal energy (E), molar capacity at constant volume (C_V_), and entropy (S) of total energy increased in the Zn_8_O_8_:4DB clusters compared with the Zn_8_O_8_ clusters due to the effect of dimethylaminobenzaldehyde on Zn_8_O_8_. The parameters are shown in Figure 9.

#### 3.6.6. Antimicrobial Activity of the Synthesized ZnO Nanoparticles ANTIBACTERIAL Assay

ZnO, ZnO:4DB NPs, and conventional antibiotic amoxicillin were tested for antibacterial activity against Gram-positive (*S. aureus* and B. subtilis) and Gram-negative (*K. pneumoniae* and *E. coli*) bacterial strains using the agar well diffusion method, as shown in Figure 10 and Figure 11. The ZnO NP, ZnO:4DB NP, and conventional antibiotic amoxicillin samples exhibited antibacterial activity. The ZnO:4DB NPs exhibited more potent antibacterial activity than the ZnO NPs. However, as the concentration of ZnO NPs increased, so did their ability to kill both Gram-positive and Gram-negative bacteria. The zone inhibition of bacterial cells may be partially due to disturbances of the cell membrane but is mainly due to their size, surface areas, oxygen vacancies, ion release, and the capacity of the reactant molecules to diffuse; nanoparticles heavily rely on the formation of active free radicals (also known as reactive oxygen species or ROS). According to the PL spectra of the present work, surface defects (oxygen vacancies for ZnO at 530 and 551 nm and ZnO:4DB NPs at 519, 530, and 560 nm) and surface charge density increased at the ZnO surface matrix with the encapsulation of 4-dimethylamino benzaldehyde compared with ZnO NPs. Nanomaterials with enhanced antibacterial activity frequently increase the production of highly reactive hydroxyl radicals (OH) by breaking water molecules inside the cell cytoplasm.

## 4. Conclusions

A straightforward chemical process was used to create ZnO NPs, employing 4DB as an organic additive. The produced particles were discovered to have a hexagonal wurtzite structure using X-ray diffraction analysis. It was found that the 4-dimethylamino benzaldehyde-added ZnO nanoparticles grew more quickly along the c-axis, which produced ZnO nanorods. According to SEM examination, the morphologies of both the pure and 4DB-added ZnO NPs indicated nanorods. EDX analysis provided an approximation of the chemical makeup of the synthesized ZnO NPs. The 4DB-added ZnO NPs’ band gap rose from 3.26 to 3.285 eV. The photoluminescence spectra revealed that all of the peaks were in the visible area and that the deep level emission (DLE) of the ZnO NPs with the addition of 4DB was also reduced. The basic set’s B3LYP/6-311 G (d, p) level was used to conduct the HOMO–LUMO analysis of the Zn_8_O_8_ and Zn_8_O_8_:4DB clusters. The samples of zinc oxide, zinc oxide:4DB nanoparticles, and amoxicillin (a common antibiotic) all displayed antibacterial activity. In comparison with ZnO, ZnO:4DB NPs displayed stronger antibacterial activity. However, when the ZnO NPs’ concentration rose, so did their capacity to eradicate both Gram-positive and Gram-negative bacteria.

## Data Availability

Not applicable.

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
