# Peer review of "Nanorod-like Structure of ZnO Nanoparticles and Zn8O8 Clusters Using 4-Dimethylamino Benzaldehyde Liquid to Study the Physicochemical and Antimicrobial Properties of Pathogenic Bacteria"

_nanomaterials, 2022, doi:10.3390/nano13010166_

Round 1
Reviewer 1 Report
The manuscript nanomaterials-2114973 requires consistent revision.
The authors mentioned that they prepared Zn8O8 and Zn8O8:4DB clusters, but they did not include the preparation method of these materials. Tha authors stated that: "The synthesized Zn8O8 and Zn8O8:4DB clusters were subjected to 26 HOMO−LUMO analysis, and their Ionization energy (I), Electron affinity (A), Global hardness (η), 27 Chemical potential (σ), Global electrophilicity index (ω), Dipole moment (μ), polarizability (αtot), 28 first-order hyperpolarizability (βtot), and other thermodynamic properties were determined." Where are these experimental data? These methods do not appear in the experimental part, nor the results in "3. Results and Discussion"!!!
These results are missing! These experimental results must be inserted in the manuscript!!!
The Figures 9a and b are missing!
The manuscript must be improved!!!
I made myself some suggestions/corrections in the attached manuscript. Authors must pay attention to the yellow and green highlighted words/sentences. Briefly, I suggested the following:
1) Lines 44-45: Reformulate the sentence!
2) Line 64: Replace "structure" with "structures".
3) Line 65: Replace "has" with "have".
4) Lines 69-70: Explain the afirmation: "These results represent the specific adsorption of carboxyl, citric acid, and phosphate ions for structural modifications." Reformulate or delete it!
5) Lines 80-81: Reformulate the sentence!
6) Authors must replace ZnO with ZnO NPs in the whole manuscript (e.g., Lines 82, 83, 350, etc.)
7) Line 89: Delete "and".
8) Lines 94-98: These experiments are missing. They must be included!
9) Line 112: The correct formula of zinc acetate is: "Zn(CH3COO)2".
10) Line 121: Specify these two pure samples! Please replace "solutions" with "suspensions". Replace "ZnO" with "ZnO NPs".
11) Line 124: Replace "ZnO solutions" with "ZnO NPs suspensions ".
Replace "solutions" with "suspensions".
12) Line 125: Replace "solution" with "suspension".
13) Line 128: Replace "Figure 1" with "Scheme 1".
14) Line 140: Replace "spectrometry" with "spectrometer". Give the name of the apparatus!
15) Line 175: Replace "ZnO" with "ZnO NPs".
16) Line 185: Replace: "is calculated by" with "was calculated with".
17) Attention to the format! See pages: 5, 6, 7, 8, 13-18.
18) Line 194: Replace "X-ray powder diffraction patterns of the ZnO NPs and ZnO:4DB NPs" with "X-ray diffraction patterns of the ZnO NPs and ZnO:4DB NPs powders.".
19) Line 211: Figure 3: The authors must delete "(a)" and "(b)" from the upper right on EDX spectra images.
20) Line 269: Where are the Figures 9a and b? The authors must insert these figures in the manuscript!
21) Line 326: Replace "OH" with "·OH".
22) Lines 331-333: The diameter of inhibition zone have mm units, not nm!!! Revise it!
23) Where are the characterization (spectral, antimicrobial, etc.) of Zn8O8 and Zn8O8:4DB clusters?
24) Line 394: Replace "partciles" with "particles".

Author Response
Response to Reviewers
Manuscript reference no: 2114973
Nanorod-like structure of ZnO nanoparticles and Zn8O8 clusters using 4-dimethylamino benzaldehyde liquid to study the physico-chemical and antimicrobial properties of pathogenic bacteria
We would like to thank Reviewers for the most appreciated comments and suggestions. We have accepted all the suggestions and we have answered all the clarifications and responses. Please find the detailed answers and the revised manuscript in which all required changes are marked in green colour. We hope that the manuscript is now substantially improved and acceptable for publication in the Nanomaterials (MDPI).
Reviewer #1:
Q#1. The authors mentioned that they prepared Zn8O8 and Zn8O8:4DB clusters, but they did not include the preparation method of these materials. Tha authors stated that: "The synthesized Zn8O8 and Zn8O8:4DB clusters were subjected to 26 HOMO−LUMO analysis, and their Ionization energy (I), Electron affinity (A), Global hardness (η), 27 Chemical potential (σ), Global electrophilicity index (ω), Dipole moment (μ), polarizability (αtot), 28 first-order hyperpolarizability (βtot), and other thermodynamic properties were determined." Where are these experimental data? These methods do not appear in the experimental part, nor the results in "3. Results and Discussion"!!!
These results are missing! These experimental results must be inserted in the manuscript!!!
The Figures 9a and b are missing!
The manuscript must be improved!!!
Response to reviewers: The potential reviewers suggested the above required results and modification is provided in the revised manuscript.
I made myself some suggestions/corrections in the attached manuscript. Authors must pay attention to the yellow and green highlighted words/sentences. Briefly, I suggested the following:
Q#2. Lines 44-45: Reformulate the sentence!
Response to reviewers: The reviewer suggested lines 44-45 is modified in the revised manuscript.
Q#3. Line 64: Replace "structure" with "structures".
Response to reviewers: Accordingly, the line 64 is modified in the revised manuscript.
Q#4. Line 65: Replace "has" with "have".
Response to reviewers: The potential reviewers indicate that the line 65 is replace the word to modified in the revised manuscript.
Q#5. Lines 69-70: Explain the afirmation: "These results represent the specific adsorption of carboxyl, citric acid, and phosphate ions for structural modifications." Reformulate or delete it!
Response to reviewers: The reviewers suggested the line 69-70 is modified accordingly in the revised manuscript.
Q#6. Lines 80-81: Reformulate the sentence!
Response to reviewers: The potential reviewers suggested line 80-81 is modified in the revised manuscript.
Q#7. Authors must replace ZnO with ZnO NPs in the whole manuscript (e.g., Lines 82, 83, 350, etc.)
Response to reviewers: Accordingly modified the above replaced the word in whole manuscript.
Q#8.Line 89: Delete "and".
Response to reviewers: The potential reviewers suggested the word is deleted in the revised manuscript.
Q#9. Lines 94-98: These experiments are missing. They must be included!
Response to reviewers: The detailed synthesis procedure is shown in the revised manuscript.
Q#10. Line 112: The correct formula of zinc acetate is: "Zn(CH3COO)2".
Response to reviewers: The required formula is Zn(CH3COO)2 is correct it in the revised manuscript.
Q#11. Line 121: Specify these two pure samples! Please replace "solutions" with "suspensions". Replace "ZnO" with "ZnO NPs".
Response to reviewers: Accordingly, the reviewers indicated that the above words are correct it in the whole manuscript.
Q#12. Line 124: Replace "ZnO solutions" with "ZnO NPs suspensions ".
Response to reviewers: The above line is corrected in the revised manuscript.
Q#13. Replace "solutions" with "suspensions".
Response to reviewers: The word is corrected in the revised manuscript.
Q#14. Line 125: Replace "solution" with "suspension".
Response to reviewers: The reviewer’s suggested word is modified in the revised manuscript.
Q#15. Line 128: Replace "Figure 1" with "Scheme 1".
Response to reviewers: Accordingly, the potential reviewers suggested the above Scheme 1 is modified in the revised manuscript.
Q#16. Line 140: Replace "spectrometry" with "spectrometer". Give the name of the apparatus!
Response to reviewers: The above word is corrected in the revised manuscript.
Q#17. Line 175: Replace "ZnO" with "ZnO NPs".
Response to reviewers: The above line is modified in the revised manuscript.
Q#18. Line 185: Replace: "is calculated by" with "was calculated with".
Response to reviewers: The required modification is done in the revised manuscript.
Q#19. Attention to the format! See pages: 5, 6, 7, 8, 13-18.
Response to reviewers: The above-mentioned pages 5, 6, 7, 8, 13-18 are corrected in the revised manuscript.
Q#20. Line 194: Replace "X-ray powder diffraction patterns of the ZnO NPs and ZnO:4DB NPs" with "X-ray diffraction patterns of the ZnO NPs and ZnO:4DB NPs powders.".
Response to reviewers: The above correction is done in the revised manuscript.
Q#21. Line 211: Figure 3: The authors must delete "(a)" and "(b)” from the upper right on EDX spectra images.
Response to reviewers: The reviewers suggested figs "(a)" and "(b)” EDX images modified in the revised manuscript.
Q#22. Line 269: Where are the Figures 9a and b? The authors must insert these figures in the manuscript!
Response to reviewers: The above-mentioned figs are corrected in the revised manuscript.
Q#23. Line 326: Replace "OH" with "·OH".
Response to reviewers: The above word is replaced accordingly reviewers suggested in revised manuscript.
Q#24. Lines 331-333: The diameter of inhibition zone have mm units, not nm!!! Revise it!
Response to reviewers: Thank you very much for your valuable comment on the above units.
The required changes are done in the revised manuscript.
Q#25. Where are the characterization (spectral, antimicrobial, etc.) of Zn8O8 and Zn8O8:4DB clusters?
Response to reviewers:
24) Line 394: Replace "particles" with "particles".
Response to reviewers: line 394 is modified accordingly suggested word in the revised manuscript.

Reviewer 2 Report
I enjoyed reading this paper because of my interest in ZnO. The authors synthesized ZnO nanoparticles using 4-dimethylaminobenzaldehyde as an organic additive. The antimicrobial application of ZnO nanoparticles was demonstrated. The synthesized ZnO nanoparticles were characterized spectroscopically and computationally, and readers of Nanomaterials will surely be interested in their findings. Therefore, this manuscript should be accepted by Nanomaterials. However, there are some problems. The authors are requested to address the following points
1) In Abstract and Conclusions, it is stated that chemical potential (σ), global electrophilicity index (ω), dipole moment (μ), polarizability (αtot), first-order hyperpolarizability (βtot), etc. were determined. However, I could not find those data in the main text.
2) The reason why the band gap of ZnO nanoparticles increases with the addition of 4DB is given on page 8. However, I could not agree with it.
The addition of 4DB decreases the size of the nanoparticles and thus increases the surface ratio. Since the coordination number of atoms on the surface is smaller, it says that the highest valence band energy will increase and the lowest unoccupied conduction band energy will decrease. If the highest valence band energy increases and the lowest unoccupied conduction band energy decreases, wouldn't the band gap become rather small?
3) Please reconsider the coloring of Figure 5. It is difficult to distinguish the many green lines.
4) Why was the cluster Zn8O8 calculated as a model? How was the size of this cluster determined?
5) Is the adsorption structure of 4DB on the ZnO cluster shown in Figure 6 stable, as it appears that adsorption is achieved by the interaction of the hydrogen atom of 4DB with the Zn atom of the cluster. I have seen many structures of organic molecules adsorbed on the surface of oxides, but I have never seen such a structure, which makes me wonder.
For example, please cite the following papers and discuss the differences in adsorption morphology.
ACS Omega 2021, 6, 34173.
Hydrogen atoms of organic molecules usually interact with oxygen atoms on the oxide surface.
Author Response
Reviewer #2:
I enjoyed reading this paper because of my interest in ZnO. The authors synthesized ZnO nanoparticles using 4-dimethylaminobenzaldehyde as an organic additive. The antimicrobial application of ZnO nanoparticles was demonstrated. The synthesized ZnO nanoparticles were characterized spectroscopically and computationally, and readers of Nanomaterials will surely be interested in their findings. Therefore, this manuscript should be accepted by Nanomaterials. However, there are some problems. The authors are requested to address the following points.
Response to reviewers: First of all, thank you very much for your valuable comments and suggestion on our manuscript. The potential reviewers suggested modifications are shown in the revised manuscript.
Q#1. In Abstract and Conclusions, it is stated that chemical potential (σ), global electrophilicity index (ω), dipole moment (μ), polarizability (αtot), first-order hyperpolarizability (βtot), etc. were determined. However, I could not find those data in the main text.
Response to reviewers: Thank you for valuable comments on the chemical potential (σ), global electrophilicity index (ω), dipole moment (μ), polarizability (αtot), first-order hyperpolarizability (βtot) results are shown in Supplementary Information.
Q#2. The reason why the band gap of ZnO nanoparticles increases with the addition of 4DB is given on page 8. However, I could not agree with it. The addition of 4DB decreases the size of the nanoparticles and thus increases the surface ratio. Since the coordination number of atoms on the surface is smaller, it says that the highest valence band energy will increase and the lowest unoccupied conduction band energy will decrease. If the highest valence band energy increases and the lowest unoccupied conduction band energy decreases, wouldn't the band gap become rather small?
Response to reviewers: Thank you very much for your valuable comments. We modified the concept of the UV band gap in the revised manuscript.
Q#3. Please reconsider the coloring of Figure 5. It is difficult to distinguish the many green lines.
Response to reviewers: The potential reviewers suggested the figure 5 is modified accordingly in the revised manuscript.
Q#4. Why was the cluster Zn8O8 calculated as a model? How was the size of this cluster determined?
Response to reviewers: Cluster Zn8O8 is calculated as a model B3LYP/6-311 G (d, p) base set using Gaussian 09W. It is a theoretical modal; we do not measure the size of the cluster.
Q#5. Is the adsorption structure of 4DB on the ZnO cluster shown in Figure 6 stable, as it appears that adsorption is achieved by the interaction of the hydrogen atom of 4DB with the Zn atom of the cluster? I have seen many structures of organic molecules adsorbed on the surface of oxides, but I have never seen such a structure, which makes me wonder. For example, please cite the following papers and discuss the differences in adsorption morphology. ACS Omega 2021, 6, 34173.Hydrogen atoms of organic molecules usually interact with oxygen atoms on the oxide surface.
Response to reviewers: Thank you very much for your valuable suggestions on the interaction of organic molecules with metal oxide clusters formation. The potential reviewers mentioned the adsorption morphologies reports are cited in the revised manuscript.
Naoaki Tsurumi, Yuta Tsuji, Noriyuki Masago, and Kazunari Yoshizawa. Elucidation of Adhesive Interaction between the Epoxy Molding Compound and Cu Lead Frames. ACS Omega 2021, 6, 34173−34184.
Shu, M.-F.; Tseng, Y.-H. Copper oxidation effect in the EMC/ Cu interfacial adhesion improvement for a novel copper interconnection substrate application. Int. Symp. Microelectron. 2018, 2018, 000161−000166.
Lee, J. H.; Kang, S. G.; Choe, Y.; Lee, S. G. Mechanism of adhesion of the diglycidyl ether of bisphenol A (DGEBA) to the Fe(100) surface. Compos. Sci. Technol. 2016, 126, 9−16.
Ogata, S.; Uranagase, M. Unveiling the chemical reactions involved in moisture-induced weakening of adhesion between aluminum and epoxy resin. J. Phys. Chem. C 2018, 122, 17748−17755.

Round 2
Reviewer 1 Report
The manuscript nanomaterials-2114973 requires minor revision.
I have corrected the manuscript again and I have attached it with the corrections yellow highlighted. I noticed that the authors did not take into account all my observations and corrections from the first revision.
1) Authors must refer to the supplementary material in the main text.
2) Attention to the font size! See the Lines: 174; 194 (Table 1; Table 2); and the relations: (1), (2), and (3).
3) Replace ZnO with ZnONPs in the manuscript: Lines: 71; 98; 194; 206; 208; 228; 241; 292; 295; 328.
4) Line 44: Replace "These" with "The".
5) Line 45: Replace "properties" with "agents".
6) Lines 45-46: Delete "by different chemical modifications".
7) Line 47: Replace "plays" with "playing".
8) Line 81: Replace "its" with "their".
9) Line 82: Replace "filed of biological studies" with "biological field".
10) Line 89: Delete "and".
11) Line 99: Latin names (e.g., "via" ) must be written italic!!!
12) Line 104: Replace "Gram positive" with "Gram-positive".
13) Line 111: Delete "of".
14) Line 181: Delete "were".
15) Line 184: Replace "is calculated by" with "was calculated with".
16) Line 210: Figure 3: The authors must delete "(a)" and "(b)" from the upper right on EDX spectra images.
17) Lines 226-227: Where is this relation? Why the authors deleted this relation? It must be inserted again!
18) Line 237: Replace "Figures 8a & b" with "Figures 5a & b".
19) Line 241: Replace "NBE" with "near band edge (NBE)".
20) Line 257: Replace "Figures 9a & b" with "Figures 6a & b".
21) Line 312: The diameter of inhibition zone must have mm units, not nm!!!

Author Response
Response to Reviewers
Manuscript reference no: 2114973
Nanorod-like structure of ZnO nanoparticles and Zn8O8 clusters using 4-dimethylamino benzaldehyde liquid to study the physico-chemical and antimicrobial properties of pathogenic bacteria
First of all, I want to express my gratitude for your thoughtful remarks and recommendations. All necessary comments and ideas have been accepted. Please refer to the amended manuscript's required revisions, which are highlighted in green, for the complete answers. Professional proof readers edit the English in the revised document. The updated manuscript should be more effective and suitable for publication in the Nanomaterials (MDPI).
Reviewer #1:
We appreciate your thoughtful comments on our text. The following suggestions are approved and changed in accordance with the updated manuscript. The amended document includes corrections for all necessary figures, tables, and references.
Q#1. Authors must refer to the supplementary material in the main text.
Response to reviewers: Potential reviewers therefore advised that the supplemental data be incorporated into the main text.
Q#2. Attention to the font size! See the Lines: 174; 194 (Table 1; Table 2); and the relations: (1), (2), and (3).
Response to reviewers: The required font size is modified accordingly in the revised manuscript
Q#3. Replace ZnO with ZnONPs in the manuscript: Lines: 71; 98; 194; 206; 208; 228; 241; 292; 295; 328.
Response to reviewers: In the amended manuscript, the words in the aforementioned lines have been changed.
Q#4. Line 44: Replace "These" with "The".
Response to reviewers: In the rewritten draft, the aforementioned word is changed to write. The updated manuscript contains all the necessary English corrections.
Q#5. Line 45: Replace "properties" with "agents".
Response to reviewers: The potential reviewers suggested the above word is replaced in the revised manuscript.
Q#6. Lines 45-46: Delete "by different chemical modifications".
Response to reviewers: The above modification is done in the revised manuscript.
Q#7. Line 47: Replace "plays" with "playing".
Response to reviewers: The above word is replaced accordingly your suggestion in the revised manuscript.
Q#4. Line 81: Replace "its" with "their".
Response to reviewers: The above word is modified in the revised manuscript.
Q#8. Line 82: Replace "filed of biological studies" with "biological field".
Response to reviewers: The above word is replaced in the revised manuscript.
Q#9. Line 89: Delete "and".
Response to reviewers: The above word is deleted in the revised manuscript.
Q#10. Line 99: Latin names (e.g., "via" ) must be written italic!!!
Response to reviewers: The above Latin names are corrected in the revised manuscript.
Q#11. Line 104: Replace "Gram positive" with "Gram-positive".
Response to reviewers: The above word is replaced in the revised manuscript.
Q#12. Line 111: Delete "of".
Response to reviewers: The above word is deleted in the revised manuscript.
Q#13. Line 181: Delete "were".
Response to reviewers: The above word is deleted in the revised manuscript.
Q#14. Line 184: Replace "is calculated by" with "was calculated with".
Response to reviewers: The above word is corrected in the revised manuscript.
Q#15. Line 210: Figure 3: The authors must delete "(a)" and "(b)" from the upper right on EDX
spectra images.
Response to reviewers: Thank for your valuable suggestion and accordingly modified in the revised manuscript.
Q#16. Lines 226-227: Where is this relation? Why the authors deleted this relation? It must be inserted again!
Response to reviewers: The above relation is inserted in the revised manuscript.
Q#17. Line 237: Replace "Figures 8a & b" with "Figures 5a & b".
Response to reviewers: Accordingly revised the above figure number in the manuscript.
Q#18. Line 241: Replace "NBE" with "near band edge (NBE)".
Response to reviewers: The above word is replaced in the revised manuscript.
Q#19. Line 257: Replace "Figures 9a & b" with "Figures 6a & b".
Response to reviewers: The above figure number is corrected in the revised manuscript.
Q#20. Line 312: The diameter of inhibition zone must have mm units, not nm!!!
Response to reviewers: The above-mentioned diameter of inhabitation zone size is modified (mm) units in the revised manuscript.
